# Amyotrophic Lateral Sclerosis in Long-COVID Scenario and the Therapeutic Potential of the Purinergic System in Neuromodulation

**DOI:** 10.3390/brainsci14020180

**Published:** 2024-02-16

**Authors:** Júlia Leão Batista Simões, Samantha Webler Eichler, Maria Luíza Raitz Siqueira, Geórgia de Carvalho Braga, Margarete Dulce Bagatini

**Affiliations:** 1Medical School, Federal University of Fronteira Sul, Chapecó 89815-899, SC, Brazil; julialeaobatistasimoes@gmail.com (J.L.B.S.); sahwebler@gmail.com (S.W.E.); siqueirarmalu@gmail.com (M.L.R.S.); braga.georgia18@gmail.com (G.d.C.B.); 2Graduate Program in Medical Sciences, Federal University of Fronteira Sul, Chapecó 89815-899, SC, Brazil

**Keywords:** amyotrophic lateral sclerosis, COVID-19, purinergic, therapy

## Abstract

Amyotrophic lateral sclerosis (ALS) involves the degeneration of motor neurons and debilitating and possibly fatal symptoms. The COVID-19 pandemic directly affected the quality of life of this group, and the SARS-CoV-2 infection accelerated the present neuroinflammatory process. Furthermore, studies indicate that the infection may have led to the development of the pathology. Thus, the scenario after this pandemic presents “long-lasting COVID” as a disease that affects people who have been infected. From this perspective, studying the pathophysiology behind ALS associated with SARS-CoV-2 infection and possible supporting therapies becomes necessary when we understand the impact on the quality of life of these patients. Thus, the purinergic system was trained to demonstrate how its modulation can add to the treatment, reduce disease progression, and result in better prognoses. From our studies, we highlight the P2X7, P2X4, and A2AR receptors and how their activity can directly influence the ALS pathway.

## 1. Introduction

Amyotrophic lateral sclerosis (ALS) is a progressive, debilitating, and fatal neurodegenerative disease of unknown etiology [1,2]. The disease mainly affects the motor cortex, brainstem nuclei, and anterior horn cells of the spinal cord [1,2] through the degeneration of motor neurons [2]. The classic symptoms, in this sense, are mainly related to muscle weakness, spasticity [2], progressive muscular atrophy [3], respiratory failure, and dysphagia [2]. However, there is evidence that behavioral and language changes are common [1]. Still, at this juncture, the typical age at which symptoms begin varies according to the etiology. Sporadic ALS usually appears around the age of 58 to 63, while familial ALS usually appears earlier, between the ages of 40 and 60 [1].

In this context, the differential diagnoses of ALS include, above all, other neurodegenerative diseases such as myasthenia gravis, Guillain–Barré syndrome, Miller Fisher syndrome, and autoimmune demyelinating disorders. Among the etiologies, the most frequently observed types are ALS of spinal origin, which most often presents with distal symptoms of weakness, and ALS of bulbar origin, which, although less frequent, is characterized by symptoms such as dysphagia and dysphonia [1]. Therefore, the severity of this disease is notable, highlighting the importance of studying its pathophysiological causes and associated neuronal changes.

On the other hand, it is necessary to highlight that infection by SARS-CoV-2—the virus that causes COVID-19—was a relevant recent cause of mortality in the world, especially in the elderly [2]. The disease is widely known to be related to typical respiratory symptoms, such as pneumonia and lung failure [2,4]. However, it is currently known that the complications of COVID are not restricted to the respiratory tract but also include events such as loss of smell and taste, headache, dizziness, stroke, and encephalitis, which occur due to the potential neuroinvasive virus [5]. Given this, it is suggested that there is a need to determine whether there is any relationship between neurodegenerative diseases, such as ALS, and virus infection.

Furthermore, COVID-19 can be associated with the persistence of a broad spectrum of symptoms and complications, even after “cure” [6]. These symptoms and complications can persist even in patients who have been infected for more than four weeks. It is interesting to highlight ringing in the ears, vertigo, dizziness [6], headaches, loss of smell, cerebrovascular events, dementia-like syndromes [4], stroke [2], and encephalitis. These complications are known as “long-COVID” or “post-COVID” [6]. Studies show, from this perspective, that 80% of people who recovered from COVID-19 experienced one or more symptoms for up to 16 weeks after recovery [6]. Therefore, it is essential to understand how post-COVID syndrome works to better understand its long-term consequences.

Among these symptoms, it is essential, therefore, to mention the impact of post-COVID syndrome on the neurological system, which can lead to cognitive impairment and syndromes similar to dementia [2,4]. Long-term neuropsychological impairments, such as attention and memory deficits, are observed even after mild infections [4]. Several recent studies, therefore, attempt to understand the pathophysiology behind these signs and symptoms, trying to relate, for example, the rupture of the blood–brain barrier (BBB) to the virus [4], the cytokine storm to neuronal inflammation [5], and the role of these in progressive neuroinflammation [7].

From this perspective, the risks of developing many neurological diseases, such as Alzheimer’s disease (AD), ALS [2], multiple sclerosis (MS) [2,8], Parkinson’s disease (PD) [4], and Huntington’s disease [5], are being associated with COVID-19 infection. However, research often reaches different conclusions about the potential risk of SARS-CoV-2 in developing these illnesses. As this is a recent situation, there is still a lack of studies that effectively understand this relationship, although some case reports already exist [1]. Therefore, it is essential to encourage more studies and analyses to be carried out to understand the action of COVID-19 in the development of neurodegenerative diseases, especially ALS.

Therefore, ALS is a severe and fatal disease that compromises the lives of those affected. COVID-19, in turn, in addition to causing characteristic respiratory symptoms, is also associated with neurological symptoms, which present risks, in particular, for neuroinflammation. Neuroinflammation, finally, is an essential factor in triggering ALS. At this juncture, we seek to understand the potential risk of COVID-19 for the development of this neurological disease, seeking to understand everything from the pathways through which neuroinflammation is generated, how it can trigger ALS, and how SARS-CoV-2 may contribute to the development of this disease in infected patients.

## 2. Methods

A narrative review study was developed to evaluate the aspects enrolled in the association between ALS and COVID-19, especially in “long-COVID”. In this regard, both the neurological consequences of the infection by this coronavirus and the potential impacts of the vaccines present in the scientific literature were explored. Furthermore, the scientific literature on the role of purinergic signaling in ALS and on the neuroprotective effects of purinergic modulation has also been evaluated.

To collect the most critical data on the delimited theme, analyze the present information in the area, and verify the proposed hypothesis, the reviewers have sought research articles in central databases, such as PubMed and ScienceDirect. In this stage, the articles included in this study were the ones related to (1) amyotrophic lateral sclerosis, (2) COVID-19, and (3) the purinergic system, which was most relevant for the research, and either evaluated the characteristics of ALS or/and SARS-CoV-2 and their associations or described the activities and role of the purinergic system in these diseases and other neurological conditions. Furthermore, interventional studies, case reports, and reviews were included.

After that, all collected data concerning the proposed hypotheses were analyzed to elucidate the possible connection between ALS, COVID-19, and the therapeutic potential of purinergic signaling neuromodulation.

## 3. The Correlation of Amyotrophic Lateral Sclerosis with SARS-CoV-2 Infection and Vaccine Implications

COVID-19, caused by coronavirus 2, has resulted in a significant pandemic [2]. In addition to its ability to cause typical, popularly recognized respiratory symptoms, evidence also proves the presence of damage to the central nervous system (CNS), which may be associated with neuroinflammation [2,4]. These damages include decreased brain size, cognitive decline, damage to brain regions related to smell, and anosmia [2,7,8]. In this sense, these impacts on the CNS and neuroinflammation raise concern about the potential long-term effects of coronavirus, especially on neurodegenerative diseases such as ALS.

Furthermore, due to the pandemic emergency, effective vaccines were developed based on well-defined clinical trials to stop the spread of the disease and reduce the intensity of symptoms in infected patients. However, even if vaccines are safe for most people, monitoring their evolution in the population and their side effects, especially for people with special medical conditions, is necessary. Therefore, since any immunizer or medication is subject to adverse situations, these must be monitored and documented [8].

From this perspective, the most common effects, such as headaches and fatigue, tend to be mild and temporary. However, it is essential to be aware of more serious charges such as anaphylaxis, Guillain–Barré syndrome, immune thrombocytopenia, and myocarditis. It is essential to be alert to signs and symptoms post-vaccination and map any physiological changes that may be found during this period to document adverse effects. In this way, it will be possible to study whether there is a relationship between immunization for COVID-19 and the development or exacerbation of symptoms of different diseases, such as neurodegenerative diseases, especially in those already predisposed to them [7].

Firstly, it will be essential to understand better the pathophysiological causes of ALS and how neuroinflammation and oxidative stress can contribute to its development in the nervous system (NS). In this regard, it will be necessary to better specify the pathways through which neuronal inflammation can be triggered [3,5,7]. Secondly, it is essential to establish how SARS-CoV-2 can encourage neuroinflammatory reactions in the nervous system [4]. Finally, more detail will be given about what new research currently offers regarding COVID-19, vaccination, and ALS [1,2,8].

It is essential to point out, at the outset, that the exact pathophysiological origin of the development of ALS still needs to be well defined [2]. However, some pathways and mediators are already well determined [3]. In this aspect, it is known that neuroinflammation, the accumulation of neurofilaments and protein aggregates, oxidative stress, glutamate excitotoxicity, and mitochondrial dysfunction may be involved in ALS. Microglia, cells similar to macrophages with regulatory and protective functions in the nervous system, can play an important role in neuroinflammation and oxidative stress, which are associated with dysfunction in motor nerve cells [3].

Some microglia can, for example, express the superoxide dismutase 1 (SOD 1) gene, which may be involved in the death of motor neurons. Several mediators and pathways may be involved in signaling for microglial activation, proliferation, and inflammation. Among these, it is worth mentioning that there are inflammatory ones, including purinergic signaling, NF-κB signaling, ROCK signaling, MAPK signaling, classical complement system, and K+ channels activated by Ca^2+^, IL-1β, TNF-α, IFN-γ, iNOS, COX2, IL-6, NO, CD86, CD68, CD14, TLR, ROS, p38 MAPK, ATP, P2X7, P2X7R, P2Y12R, P2Y13R, P2Y14R, NF-κB, NAIP, TDP-43, Ca^2+^, and K^+^, and anti-inflammatory ones, including fractalkine signaling, TREM2 signaling, TNF-α, IL-1β, IL-6, iNOS, APOE, and CX3CR1 [3,5,7].

In this circumstance, microglia can manifest themselves in two states: the so-called M1, with a proinflammatory characteristic, and the M2, an alternative state with an anti-inflammatory characteristic [3]. M1 microglia, thus, express molecules and cytokines that include tumor necrosis factor-α (TNF-α), interleukin-1β (IL-1β), interferon-γ (IFN-γ), and nitric oxide (NO) [3]. In addition to these molecules, other inflammatory cytokines may have their production stimulated at this juncture. They may remain elevated even after healing, such as IL-6, IL-8, IL-10, IL-16, IL-17A, and IL-18, which generate the so-called cytokine storm commonly found in COVID-19 [5].

M2 microglia, especially those induced by interleukin-4 (IL-4), express other molecules, such as IL-4 itself, arignase1, Ym1, CD206, and IL-10, thus presenting neuroprotective effects [3]. From this perspective, it is notable that understanding this double manifestation that microglia can perform [3] and understanding the context of the cytokine storm associated with COVID, based on, for example, the analysis of biomarkers from cerebrospinal fluid (CSF) [5], is crucial for the early detection of the exacerbation of neurodegenerative diseases associated with coronavirus 2. Thus, it will be possible to better describe the mechanisms behind neuronal inflammation and oxidative stress, some of the likely causes of ALS [2,3].

In this sense, one of the main neuroinflammatory signaling pathways is from the receptor expressed on myeloid cell 2 (TREM2) [7], which is a protein linked to the immune system membrane expressed exclusively by CNS microglia. Although studies in animal and human models regarding the activity of this protein differ in the exact action of this molecule, it cannot be ignored that it represents a possible new therapeutic target for treatments. This receptor, for example, when interacting with molecules such as TDP-43, demonstrates relevant anti-inflammatory potential. Furthermore, research indicates that TREM2 can also induce APOE signaling, preventing microglial neuronal degeneration [3,7].

Furthermore, it is necessary to mention the performance of the purinergic system, which can act as a modulation pathway for neuropathic pain and a signaling system for microglial behavior in the CNS. Purinergic signaling involves nucleotides such as Ado, ATP, ADP, AMP, ionotropic receptors (P2X), purine nucleotide receptors (P2Y), P1, P2 receptors, and extracellular enzymes, which are often stimulated by ATP [3]. Thus, these receptors may represent drug targets for inhibition and stimulation to control the consequences of neurodegenerative diseases, such as pain.

The P2X7R receptor, for example, which has a low affinity in microglia, has a vital inflammatory role when sensitized by ATP. Extracellular nucleotides, in turn, play a crucial role in communication between neurons and microglia through purinergic P2X and P2Y receptors expressed on microglia, especially in situations such as neuropathic pain after nerve injuries, in which microglia increase the expression of purinergic receptors. Furthermore, P2Y13R/P2Y12R receptors contribute to developing neuronal hyperexcitability, a phenomenon associated with neuropathic pain. In the context of neuropathic pain, it is also worth noting that pharmacological blockade of the P2X4R receptor reduces pain-related behaviors in animals, indicating its potential as a therapeutic target [3].

On the other hand, the Rho/ROCK pathway, which comprises ROCK 1 and ROCK 2, plays a crucial role in several physiological activities. The ROCK1 pathway is predominant in the liver, lungs, testicles, blood, and immune system, while the ROCK2 pathway is dominant in the brain and muscles. Therefore, in the context of ALS, the main active pathway is ROCK2, and it is essential to understand its inflammatory performance in regulating microglial activity and inflammation. In this sense, ROCK inhibitors, such as fasudil, show therapeutic potential in ALS by modulating microglial function and reducing the release of proinflammatory factors [3].

Another relevant signaling method in the context of ALS is signaling via mitogen-activated protein kinase (MAPK), which is directly associated with oxidative stress and neurodegeneration [3,7]. The overactivation of this protein leads to the uncontrolled release of cytokines, contributing to the generation of neuronal damage. Several studies relate the widespread presence of MAPK to different neurodegenerative diseases, such as AD, PD, MS, and ALS [7]. At this juncture, MAPK inhibitors such as SB203580 and compounds such as sapogenin demonstrate promising anti-inflammatory effects, but more research is needed for future clinical development [3].

Fractalkine, in turn, plays a crucial role in regulating the microglial response in conditions such as ALS, as it acts as a neuroprotector. Fractalkine is a chemokine with high expression in the brain, synthesized by neurons. Its protective activity acts primarily through phagosomal regulation. Research suggests that manipulating fractalkine signaling may be a potential therapeutic target for treating ALS, with additional studies needed to better understand its molecular mechanism and confirm its possible therapeutic role in humans [3].

Nuclear factor kappa-light-chain-enhancer of activated B cells (NF-κB) also plays a crucial role in several neurodegenerative diseases, including ALS and SMA, activated in the spinal cord, directly regulating inflammation [3,7]. In ALS, activation of the NF-κB signaling pathway is associated with neuroinflammation [7], with activated microglia contributing to the increase in proinflammatory cytokines. Studies reveal that NF-κB activation triggers the transformation of microglia to a proinflammatory phenotype (M1), characterized by elevated expression of proinflammatory genes. Intervention of NF-κB shows therapeutic promise with the inhibition of this pathway, either through specific inhibitors or drugs such as minocycline, demonstrating the rescue of motor neuron survival in ALS models [3].

Furthermore, the Ca^2+^-activated K+ channel (KCa3.1), expressed by microglia, regulates cell migration and phagocytic activity. Studies indicate that the activation of purinoceptors such as P2X4 and P2X7 is related to the production of TNF-α by microglia, activating Ca²-dependent responses to external stimuli. Thus, there is increased expression of proinflammatory genes and a reduction in anti-inflammatory genes, a scenario commonly found in ALS patients. This scenario also highlights a possible therapeutic response that aims to act on these receptors [3].

After assimilating the signaling pathways associated with neuroinflammation and neuronal degeneration [3], it is essential to understand SARS-CoV-2’s ability to contribute to this neurodegeneration [7]. This pathogen can directly invade the brain through the olfactory bulb, retrograde axonal transport, peripheral nerve endings, the hematogenous or lymphatic route, and mainly through the BBB [3,7]. Such neuronal infection, combined with the activation of peripheral leukocytes, results in the positive activation of the SN of inflammatory cytokines, which leads to neurodegenerative changes related to inflammation [7].

## 4. Potential Pathways in the Pathophysiology of Amyotrophic Lateral Sclerosis Associated with COVID-19

Increasing evidence has suggested that COVID-19 has the potential to cause systemic inflammation. This inflammatory process can, for example, result in sepsis, which induces hypoxemia and, consequently, neurodegeneration. From this understanding, it is noted that the trend in the post-pandemic world is an increase in cases of neurodegenerative diseases since the virus can stimulate the pathways associated with triggering these diseases [7]. Accordingly, there are already reports of patients who, even after mild infections, had a loss of memory and attention, executive deficits, and other neuropsychological impairments. From this, the hypothesis of a possible association between this virus and the possibility of developing neurodegenerative diseases was outlined (Table 1) [4].

Recent research published in 2022 using blood biomarkers of BBB rupture and neuronal damage in patients with COVID-19 found that these biomarkers are elevated in both COVID-19 and ALS patients. This study used two cohorts of patients with COVID-19, one without neurological damage and the other with patients with neurological complications, and two control groups not infected by SARS-CoV-2, one healthy group and the other with ALS. In this regard, it is believed that the pathogenic process that leads to neurodegeneration includes viral invasion of the BBB, bioenergetic failure, and autoimmune and innate neuroimmune responses, which trigger neuroinflammation [4].

Several studies try to understand, based on case analyses, for example, whether it is possible for COVID-19 and its long-term consequences or vaccination for this disease to trigger ALS and other neurodegenerative diseases [1,2,8]. Since SARS-CoV-2 can cross the BBB [4] and provoke neuroinflammatory events [3], and the pathophysiology of ALS may be related to neuroinflammation, it is possible to hypothesize that infection by this respiratory virus may ultimately result, for instance, in the development of ALS. Considering the recent pandemic, it is essential to develop broad studies on the topic and carefully analyze the information already published, aiming to cover what is currently known about the consequences of this emergency that recently shook the world.

In this scenario, two relevant studies are published that draw the parallel between SARS-CoV-2 infection and the development and progression of neurodegenerative diseases. One of these studies refers to a case report of bulbar-onset ALS after COVID [1], and the other is a comprehensive survey that used genetic data to perform genome-wide association analyses (GWASs) [2]. Furthermore, another significant article worth mentioning is the case report of a patient who developed ALS after vaccination with J&J/Janssen [8]. These studies reported no conflict of interest and provided valuable information about the possible consequences of COVID-19 [1,2,8].

One of these studies, published in 2023, followed an 84-year-old man. Initially, this man went to the emergency room with complaints of progressive dyspnea, a two-day dry cough, and bilateral tingling in the upper and lower extremities. His vital signs, at that time, were stable, with a temperature of 37.1 °C and an oxygen saturation (SpO2) of 85%, improving to 94% with oxygen therapy. Furthermore, on physical examination, he had coarse inspiratory and expiratory crackles. Given the symptoms, the COVID-19 test was carried out, and despite having received four doses of the vaccine, it presented a positive result, and, consequently, he was hospitalized [1].

Due to persistent coughing, this older man was evaluated by the speech therapy service, confirming that swallowing control was free from irregularities. After five days of hospitalization and improvement in oxygen dependence, the patient was allowed to return home. However, a month after discharge, the man noticed a worsening cough and difficulty swallowing that he had not previously seen. On physical examination, although he was well oriented, this patient presented muscle loss and dysarthria, which suggested nerve paralyzes such as the vagus, glossopharyngeal, and hypoglossal nerves. Because of this, it was decided to keep him on parenteral nutrition [1].

Even after a series of imaging tests such as computed tomography (CT) and magnetic resonance imaging (MRI), laboratory tests such as CK and vitamin levels, and CSF analyses, the diagnosis and referral to the ALS center occurred late, as, initially, these tests demonstrated little or no changes suggestive of this disease. The most relevant thing in this case, in addition to ALS appearing soon after SARS-CoV-2 infection in a man with no previous signs of the disease, is that the symptoms evolved much faster than the pattern that is generally found [1].

This article emphasizes that it is not sure whether COVID-19 was the causative element of ALS in the patient. However, it highlights that there is no way to ignore the fact that ALS presented itself soon after the infection and evolved much faster than usual. Given this, the research brings up discussion about not only ALS as a consequence of COVID-19 but also the accelerated progression of ALS as a possible effect on people already affected. The suggested mechanisms for this process include neuroinvasion, neuroinflammation, and BBB dysfunction [1].

Another relevant article, published in 2022, conducted a Mendelian randomization (MR) study on two samples to evaluate the relationship between COVID-19 and several neurodegenerative diseases. This study found a direct causal relationship between AD and COVID-19 but did not find such a relationship for ALS or MS. Although this result is in line with the suggestion given by the 2023 case report, it is essential to establish that the pandemic is still a recent event, and only time and broader research will be able to indicate, in fact, the relationship between SARS-CoV-2 and ALS. Regardless, it is already known that COVID-19 negatively affects cognitive function, as it can destabilize the BBB and trigger proinflammatory cytokines in the brain [2].

Finally, the study by Feghali et al. [8] addresses the case report of a 47-year-old man who, after immunization with the J&J/Janssen COVID-19 viral vector, developed ALS. The patient in question did not have a previous diagnosis of ALS but had a family history of the disease, which indicates a probable genetic predisposition to it. In this way, the study emphasizes, based on an isolated case report, the need to pay attention to the side effects of the immunizer in patients who are already predisposed to neurodegenerative diseases such as ALS [8].

In this sense, it is essential to clarify that there are few reports of possible serious side effects of vaccines. However, there is much evidence of their benefits in the context of the pandemic. The most recent studies even indicate that vaccination is the best preventive strategy and that the relationship between vaccination and ALS is fragile. In any case, monitoring patients already predisposed to neurodegenerative diseases from before until after immunization can help to understand the pathophysiology of these diseases and verify that this portion of the population, in fact, benefits from the use of this immunizer [8].

In this published study, it was reported that shortly after vaccination, the patient presented non-specific symptoms, such as inflammation at the injection site. However, in the same week, he began to notice weakness in that arm, which worsened and progressed to other limbs in the following months. ALS was confirmed after a referral to a neurologist. The article highlights that there is no way of knowing, in this case, whether the ALS symptoms were triggered by the vaccine, as other factors, such as the man’s propensity for the disease or the presence of symptoms and progressive worsening, may have influenced it [8].

Thus, the need for in-depth and broad research and studies that aim to thoroughly investigate the consequences of the COVID-19 pandemic on neurodegenerative diseases is evident, especially in the context of post-COVID syndrome. Current research linking SARS-CoV-2 to ALS still needs to be expanded, not allowing for a definitive conclusion. Furthermore, understanding the pathways and mediators of the development of neuroinflammation and oxidative stress in the CNS is essential for developing efficient and targeted treatments. In this way, it will be possible to outline new proposals for ALS development, develop better diagnostic methods, and understand risk factors that were previously unspecified (Figure 1).

## 5. Aiming at the Therapeutic Potential of the Purinergic System in Reducing Oxidative Stress and Neuroinflammation

Even though it is a widely debated pathology, considerable challenges remain in the development of therapies capable of modifying the course of ALS. Given this scenario, there is a growing need to identify new targets for creating drugs that can delay the onset and progression of this pathology [9,10]. In this context, the importance of the purinergic system is highlighted, and the feasibility of developing medications targeted at these receptors as a possible therapeutic approach for ALS is discussed [11].

The purinergic system plays a crucial role in the functioning of the SN, being responsible for regulating essential signaling molecules such as ATP and ADP. These substances perform a vital function in cellular communication, triggering fundamental processes for the efficient coordination and integration of neuronal activities. Dysregulation of this system is closely associated with neurodegenerative and neuroinflammatory diseases, with ALS being one of the conditions included in this spectrum. In this context, purinergic receptors have been widely studied to identify their role in this pathology and possible alternative therapeutic strategies [9,11,12].

In ALS, a significant relationship exists between the regulation of P2 receptors and a decrease in the hydrolysis capacity of extracellular ATP. In a study by D’Ambrosi et al. [13], the activity of P2X4, P2X7, and P2Y6 receptors and a simultaneous downregulation of ATP hydrolyzing activities in ALS microglia were observed. This association points to an inflammatory cycle in which ectonucleotidases are negatively modulated to preserve the ATP reservoir available for P2 receptors. This dynamic plays a significant role in the perpetuation of inflammatory mechanisms in ALS, contributing to the maintenance of the pathological process [14].

Among the P2 receptors, P2X7 stands out as the main one in this context, and several studies widely corroborate its relevance in the development of ALS. Evidence suggests that P2X7 plays a role in the initiation and progression of pathology. Activation of this receptor by 2′-3′-O-(benzoyl-benzoyl) ATP (BzATP), a P2X receptor agonist, resulted in increased production of reactive oxygen species (ROS) in mouse microglia. This effect was achieved through the activation of NADPH oxidase 2 (NOX2), an enzyme widely recognized in the pathogenesis of ALS [13,15].

Furthermore, evidence points to a significant increase in the expression of the P2X7 protein in microglia and macrophages in the spinal cord of patients diagnosed with this disease. In this regard, a study analyzing frozen human postmortem spinal cord specimens found that individuals who had ALS or multiple sclerosis (MS) presented an increased density of P2X7 receptors in the regions affected by these diseases when compared to healthy individuals, demonstrating the involvement of this receptor in ALS and other neurodegenerative diseases [16].

During the initial phase of ALS, the expression of the P2X7 protein remains constant in the motor cortex, suggesting its active participation in the pathological process. However, in peripheral blood mononuclear cells, a downregulation of the expression of this protein was observed. These findings highlight the complexity of the interaction between the P2X7 receptor and ALS, highlighting the importance of considering different contexts and tissues when exploring this receptor’s role in the disease’s pathophysiology [11].

The activation of the P2X7 receptor by inflammatory microRNAs (miRNAs) has also been investigated, revealing a negative regulation in the IL-6/STAT3 interleukin signaling pathway. Consequently, an increased production of TNF-α was observed, contributing to a detrimental microglia phenotype [17]. Therefore, it is assumed that neuroprotective agents, such as P2X7 antagonists, can interrupt the harmful effects of activating this purine receptor. Thus, it is possible to infer that such substances could delay the progressive damage of motor neurons [15]. An example of this therapeutic approach emerges with the P2X7 antagonist BBG, tested in mice at different stages of ALS. According to Apolloni et al. [18], the results acquired with the administration of BBG during the late presymptomatic phase proved promising, delaying the onset of the disease and improving the performance of motor neurons [18].

Another factor studied was the role of mutant superoxide dismutase 1 (SOD1) in the spread of ALS. In familial cases of the disease, an association with mutations in SOD1, an enzyme with antioxidant properties, was observed. The dysfunction or inappropriate aggregation of SOD1 emerges as a crucial factor in the origin and propagation of neurodegeneration, being the subject of intense investigation in several studies [19]. According to Bartlett et al. [20], research conducted in rodents revealed that the activation of P2X7 by extracellular ATP influences the release of SOD1-G93A (a mutant form of the enzyme) from NSC-34 motor neuron cells previously transfected with SOD1-G93A. These released enzymes can, in turn, be transmitted to naive NSC-34 cells, resulting in the spread of the disease [20].

Furthermore, it was found that the P2X7 protein is positively regulated in the spinal cord, especially in mice carrying SOD1-G93A, highlighting its potential implication in the development of ALS. This research suggests the possibility of therapeutic benefits by antagonizing P2X7 activity in individuals affected by ALS with the SOD1-G93A mutation [20].

Another receptor that stands out in the context of ALS is P2X4. Recent research shows that mutant SOD1 enzymes increase P2X4 density in cells that express this receptor. Additionally, an increase in P2X4 expression in microglia was observed during the symptomatic phase of the disease. A relevant aspect to be considered is that, in studies conducted with mice carrying SOD1, a positive regulation of the presence of P2X4 on the cell surface was found. This finding suggests the need to investigate whether this phenomenon also occurs in monocytes from ALS patients. This approach could provide valuable information, enabling the identification of a potential early biomarker for detecting the disease even before the manifestation of clinical symptoms [12].

In this scenario, no studies could be found regarding the elevated levels of these receptors in humans, indicating the need for further studies concerning this receptor activity in neurodegenerative diseases. Nonetheless, studies in vivo with mice have indicated an elevation in the expression of P2X4 receptors in the cells of specimens that suffered spinal cord injury in comparison with the control group [21]. Along with that, in tests with mice, P2X4 receptor knocked-out specimens presented lower neuroinflammation levels after spinal cord injury than the control group [22]. Both of these findings indicate that the presence of P2X4R activity in the nervous system can induce neuroinflammation, contributing to ALS. 

Furthermore, the role of other purinergic receptors in ALS has also been studied. According to D’Ambrosi et al. (2009), a positive expression of the P2Y6 receptor was observed in the microglia of SOD1-G93A mice. Additionally, studies with the 1321N1 astrocytoma cell line indicated that P2Y6 receptor expression avoided apoptosis when these cells were exposed to chemical ischemia, oxidative stress, and death receptor activation by TNFα [23]. In this context, P2Y6 can have a protective effect on neurodegenerative conditions, such as ALS, since it apparently is capable of reducing neuron loss. 

Likewise, the P2Y12 purinoreceptor has been the target of investigation as a potential marker of the resting/vigilance state of microglia in ALS since its expression is reduced in spinal cord microglia during the neuroinflammation process in patients with this pathology [13]. Corroborating this idea, studies with mice have found that the expression of the P2Y12 receptor is increased when the specimen is exposed to spinal cord injury, reducing its activity around 30 days after the injury, confirming its responsiveness to spinal cord damage [24]. However, both P2Y6 and P2Y12 behaviors have not been studied in vivo in humans; hence, the need for further analysis in this regard must be highlighted.

Regarding P1 receptors and adenosine markers (A1R and A2AR), investigations in SOD1-G93A mice revealed a decrease in A1R protein levels in the cortex during the presymptomatic stages of the disease. However, there was no change in the spinal cord. This early reduction may compromise the effectiveness of synaptic modulation, compromising the neuroprotective role of A1R and aggravating the neurodegenerative process [25]. In symptomatic mice, it was observed that A1R showed a more pronounced tonic activation associated with high levels of extracellular adenosine, which could result in muscle or neuronal degeneration. Thus, the role of A1R in neurodegenerative diseases, such as ALS, still lacks a more profound understanding, despite evidence that supports its neuroprotective capacity [26].

The role of A2AR in ALS is complex, as there is contradictory evidence regarding its activation and blockade effects on disease progression. The notable disparities in the neuroprotective actions of A2AR arise from the diversity of functions performed by these proteins. During the presymptomatic phase, a transient increase in A2AR levels was observed in the spinal cord, while in the initial symptomatic phase, this increase was identified in the cerebral cortex. The complex interaction between the decrease in A1R and the increase in A2AR may contribute to cortical hyperexcitability, a phenomenon reported in the early stages of motor symptoms, both in familial and sporadic cases of ALS [25].

Furthermore, a study by Liu et al. [27] highlighted that A2AR agonists showed benefits by improving the survival of motor neurons. On the other hand, in research by Potenza et al. [28], adenosine receptor antagonists, such as caffeine, had detrimental effects on the survival of mice carrying the SOD1-G93A mutation. Thus, the functional duality of A2AR in ALS highlights the complexity of the interaction between these receptors and the need for more refined approaches to understand their role in the progression of this pathology. It is clear, therefore, that the therapeutic approach with adenosine receptor ligands can lead to opposite effects due to the compensatory regulation of these receptors. There are still several discrepancies in the literature regarding the therapeutic properties of these receptors, which indicates a need for future research that addresses this topic (Figure 2) [25,26].

## 6. The Interaction of SARS-CoV-2 on Purinergic Receptor Expression

Considering the association between ALS and the purinergic system, it is possible to infer that the influence of SARS-CoV-2 on ALS may be mediated by purinergic signaling and that COVID-19 also impacts purinergic activity. In this sense, some studies suggest changes in the purinergic pathways triggered by this virus. The research by Pietrobon et al. [29] demonstrated impairment of the activity of ectonucleotidases in individuals infected by the virus by analyzing the plasma concentrations of these enzymes in the blood. They also observed reduced Ado production, with both conditions contributing to a proinflammatory state [29].

Furthermore, the work of Díaz-García et al. [30] also evaluated CD39 activity in blood samples from patients infected with COVID-19. As a result, they found the same reduction in the activity of this enzyme. However, they also discussed the role of nucleotide accumulation as a cause of the activation of other purinergic receptors. In this sense, they suggest an increased activation of the P2Y12 receptor, contributing to thrombotic episodes in patients with SARS-CoV-2. They also suggest an essential activation of the P2X7 receptor, which is responsible for mediating the proinflammatory conditions caused by the virus [30].

Other research, developed by García-Villalba et al. [31], also analyzing blood samples, corroborates the hypothesis about the activity of P2X7 in COVID-19 inflammation by finding an increased expression of soluble P2X7 receptors in the plasma of infected individuals when compared with the control group. This study also observed higher levels of soluble P2X7 in patients with more severe cases of COVID-19, with worse prognoses, indicating an association between increased activity of this P2 receptor and a more worrying outcome. Research suggests that P2X7 may also be a biomarker for inflammatory activity in SARS-CoV-2 infections [31].

Lécuyer et al. [32] correlate P2X7 and the activation of the NLRP3 inflammasome in the pathogenesis of SARS-CoV-2 infection since the activation of these inflammasomes was detected during COVID-19 and is closely associated with the severity of the condition [33]. It is understood that several inflammasomes are activated, but NLRP3 is highlighted for its contribution to organ dysfunction and disease severity during COVID-19 [33]. Thus, it is associated with the chronic stages of COVID-19 in humans [33] and preclinical mouse models [34,35] since the activation of NLRP3-dependent inflammasomes produces proinflammatory substances. Among them, the production of uncontrolled extracellular traps (NET) by neutrophils has been proposed to contribute to hyperinflammation and dysregulated coagulation in patients with severe disease [36,37,38].

Therefore, regarding the neuroinflammatory scenario, P2X4 and P2X7 receptors are ion channels regulated in these conditions [39,40,41]. These receptors produce free radicals and increase oxidative stress in inflammatory and neurodegenerative diseases [42,43]. TLR4 stimulation increases P2X7 expression in microglial cells and astrocytes [44,45], whereas P2X7 receptor inhibition decreases LPS-induced cytokine production in cultured human microglial cells and brain tissue from septic mice [43,46]. Notably, the P2X7 receptor is the second signal for activating the NLRP3 inflammasome and secretion of IL-1, a cytokine closely related to synaptic loss and depressive behavior following inflammatory and chronic stress conditions [47,48,49]. From this perspective, Alves et al. [50] correlate the increase in P2X7 receptor levels in BV2 microglial cells and in the hippocampus of mice injected with SARS-CoV-2 spike protein, suggesting that P2X7 may contribute to COVID-19-related neuroinflammatory events.

From another perspective, Rud et al. [51] analyzed plasma from patients hospitalized with COVID-19 for the effects of infection on CD73 activity. As a result, they also found a reduction in the expression of these ectonucleotidases. However, they suggested that reduced nucleotide hydrolysis leads to reduced Ado production, preventing its expression of anti-inflammatory activity [51]. Considering the indications from scientific data of the anti-inflammatory effects of the P1 receptor family, this lack of Ado can be seen as a mediator for reducing the anti-inflammatory and neuroprotective activity presented by these receptors [52].

Therefore, COVID-19 infection appears to increase the activity of P2 receptors linked to systemic inflammation, reducing the actions of ectonucleotides, impairing ATP hydrolysis, and generating its accumulation [29,30]. Additionally, it can also be inferred that this lack of hydrolysis causes a reduction in Ado concentrations, leading to a decrease in the activity of P1 receptors, impairing their protective and anti-inflammatory conditions [29,51,52]. Thus, SARS-CoV-2 can affect purinergic signaling, preventing its neuroprotective effects and contributing to systemic inflammation, which is a dangerous condition for ALS patients.

## 7. Discussion

From the various studies, it is possible to perceive the significant role of the purinergic system, especially the P2X7 and P2X4 receptors, in the context of ALS. P2X7 receptors are implicated in inflammation and the spread of disease, with studies suggesting that their activation can increase ROS production in microglia [9,12,13,14,15]. The negative regulation of P2X7 expression in different tissues indicates a complex interaction with ALS. The inhibition of this receptor, as demonstrated by the antagonist BBG in studies with mice, shows promising therapeutic perspectives, delaying the onset of the disease [11,14,18].

The P2X4 receptor is also highlighted, with research indicating an increase in its expression in microglia during the symptomatic phase of ALS. Studies suggest that P2X4 can serve as a potential biomarker for early disease detection, offering valuable information before the manifestation of clinical symptoms [12].

The interaction between adenosine receptors (A1R and A2AR) in ALS is complex, with contradictory evidence on their activation and blockade effects on disease progression. A2AR shows a functional duality, contributing to cortical hyperexcitability in the early stages of motor symptoms. However, studies suggest that A2AR agonists may benefit motor neuron survival, while antagonists like caffeine may have detrimental effects. The compensatory regulation of these receptors highlights the need for more refined approaches to understand their role in ALS [25,26,27,28].

However, until today, only a few studies have been developed concerning the role of specific A2A modulators in ALS. In this sense, one of the main studies developed is related to the effects of KW-6002, an A2A antagonist, in mice with ALS [53]. As a result, Rei and collaborators found that, when initiated early, this drug is capable of reducing the severity of ALS in the more advanced stages of the disease [53]. Nonetheless, considering the need for a delicate balance of A2A receptor activity in ALS and the lack of strong evidence on the effects of the modulation of this receptor in this scenario, the need for further studies must be highlighted. 

The notable influence of purinergic receptors on ALS becomes evident, particularly P2X7, given its specific involvement with neuroinflammatory mechanisms. These findings suggest that antagonists targeting this receptor may represent a promising perspective for the development of effective treatments, renewing hopes in the search for practical solutions for ALS [11,13,14]. In this regard, different studies have recently analyzed drugs capable of acting on P2X7 receptors with this mechanism. In 2021, scientists tested the effects of the P2X7 antagonist AXX71 in mice with ALS [54]. As a result, Apolloni et al. [54] found a reduction in neuroinflammation and ALS progression in mice that received the drug, indicating the promising effects of this P2X7 antagonist.

Furthermore, another important P2X7 antagonist analyzed in mice with ALS was JNJ-47965567 [55]. In this study, female mice that received the therapy presented a delay in the disease progression and amelioration of motor activities. However, male mice did not suffer any impact from the therapy, indicating the need for further studies regarding the potential effects of this drug and its different effects on different sexes [55]. 

Finally, it is important to mention the importance of analyzing these promising perspectives in purinergic modulation. Currently, the main drugs approved in different regions of the world for the treatment of ALS are riluzole, edaravone, and tauroursodiol/sodium phenylbutyrate [56]. Despite the positive effects of these drugs, considering the severity of ALS and the absence of a final effective therapy for this neurodegenerative disease, more pharmaceutical targets must be explored [56]. In this sense, purinergic modulation has presented very interesting results in ALS control through pathways different from the currently known drugs, which act through other pathways by promoting anti-inflammatory and anti-apoptotic effects, reduction in oxidative stress, and modulation of neuronal ion channel activities [57,58,59].

Hence, analyzing purinergic signaling is essential in the context of ALS, a disease with such a worrying prognosis. Thus, exploring purinergic modulation in this neurodegeneration is fundamental for developing drugs that are either capable of functioning as individual therapies or as adjuvants.

## 8. Conclusions

Despite the lack of research evaluating and confirming the association between ALS and SARS-CoV-2, a considerable number of studies indicate a link between both conditions. In this regard, even though the pathophysiology of ALS is not completely understood, multiple pathways triggered by the infection or contact with the virus seem to be capable of linking this neurodegeneration to COVID-19. Considering that and the infectiousness of SARS-CoV-2, it is very important to study this phenomenon carefully and elucidate the main processes involved in it.

Furthermore, the severity of ALS and the need for effective therapies for this disease must be highlighted as well. With that in mind, purinergic modulation has been associated with neuroinflammation and ALS. In addition, different studies demonstrate the positive impacts of purinergic signaling modulation in neurodegenerative conditions, making it a pharmacological target against ALS. Therefore, further research must aim at explaining the precise role of purinergic signaling in ALS and possible strategies to modulate its activities, avoiding the development and progression of this serious condition.

## Figures and Tables

**Figure 1 brainsci-14-00180-f001:**
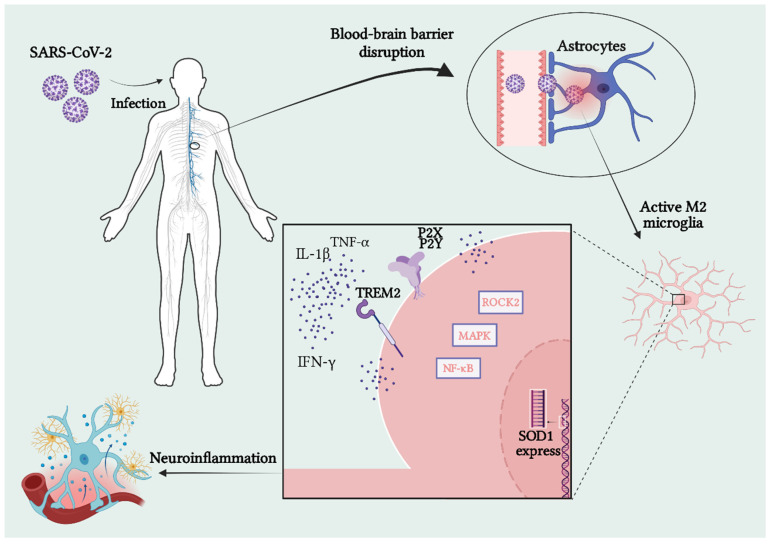
Pathways present in SARS-CoV-2 infection in the neuronal environment and their influence. Recent studies have shown that SARS-CoV-2 infection has the potential to break the BBB. This process stimulates the activation of M2 microglia, which are responsible for mediating inflammatory responses. Pathways and mediators such as the purinergic receptors P2X and PSY, the TREM2 receptor, the ROCK2, MAPK, and NF-κB pathways, together with the overexpression of the SOD1 gene, actively participate in this process through the release of essential cytokines such as IFN-γ, IL-1β, and TNF-α. This set can result in inflammation and neuronal death.

**Figure 2 brainsci-14-00180-f002:**
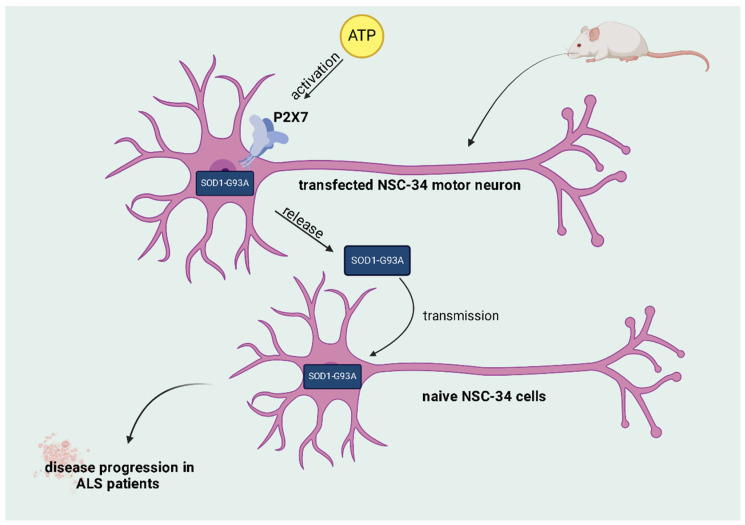
The role of SOD1 in the pathophysiology of ALS and its relationship with the purinergic system. The role played by SOD1 in the development of ALS in familial cases: activation of the P2X7 receptor by extracellular ATP influences the release of SOD1-G93A from NSC-34 motor neurons previously transfected with the mutant enzyme. The released enzymes can be transmitted to unaffected NSC-34 cells, thus contributing to the spread of the disease. This function of SOD1 suggests therapeutic benefits when using P2X7 antagonists in individuals affected by ALS who have this mutation.

**Table 1 brainsci-14-00180-t001:** Studies analyzing the association between ALS and COVID-19.

Reference	Type of Study	Type of Contact with the Virus	Association between COVID-19 and ALS
Bonetto et al., 2022 [4]	Cohort study	Infection	Associated, due to BBB damage, bioenergetic failure, and autoimmune and innate neuroimmune responses.
Zhang & Zhou, 2022 [2]	Mendelian randomization	Infection	Not associated
Abu-Abaa et al., 2023 [1]	Case report	Infection	Suggested
Feghali et al., 2023 [8]	Case report	Vaccine	Suggested

## Data Availability

The cited studies are publicly available.

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
