# Peer review of "Amyotrophic Lateral Sclerosis in Long-COVID Scenario and the Therapeutic Potential of the Purinergic System in Neuromodulation"

_brainsci, 2024, doi:10.3390/brainsci14020180_

Round 1

Reviewer 1 Report

Comments and Suggestions for Authors

The review “Amyotrophic Lateral Sclerosis in Long-COVID Scenario and the Therapeutic Potential of the Purinergic System in Neuromodulation” submitted to Brain Sciences, is a narrative study based on data retrieval from central databases such as PubMed and ScienceDirect, according to the specific: (1) amyotrophic lateral sclerosis, (2) COVID-19, and (3) purinergic system, queries (chapter 2).

The Authors in brief summarize the pathogenesis and major symptomatology of ALS, the complications of long-COVID targeting especially the nervous system and causing neurodegeneration/neuroinflammation, and the impact of the purinergic system particularly on ALS (chapter 1). Moreover, they highlight the correlations between ALS and SARS-CoV-2 infection and vaccine implications (chapter 3), finally describing the potential pathways that are associated to COVID-19 in the pathophysiology of ALS (chapter 4).

Although these specific topics are thoroughly described in several previous studies, these chapters are clear and exhaustive, despite being concise but not original.

The novelty of this review resides instead in the therapeutic potential of the purinergic system in the neuromodulation of ALS in the long-COVID scenario, as stated by the Authors in their review’s title and keywords, and well depicted in the very captivating Graphical Abstract.

However, the Authors have completely neglected to report about the abundant literature existing on the impact of purinergic signaling in the SARS-CoV-2/COVID-19 neuroinflammatory context. This is quite surprising, having the Authors themselves published several articles on this specific topic.

I would suggest adding a significant chapter describing and highlighting this particular relationship.

You may find here some of the most recent articles, just as an example:

doi: 10.3389/fimmu.2023.1158460

doi: 10.3389/fimmu.2023.1182454

doi: 10.2147/JIR.S413892

doi: 10.3389/fimmu.2023.1270081

Some copyediting and proofreading would be also valuable.

Comments on the Quality of English Language

Some copyediting and proofreading would be valuable.

Reviewer 2 Report

Comments and Suggestions for Authors

In the article entitled "Amyotrophic Lateral Sclerosis in Long-COVID Scenario and the Therapeutic Potential of the Purinergic System in Neuromodulation", the authors make a narrative review on the pathophysiology of amyotrophic lateral sclerosis associated with SARS-CoV-2 infection, exploring therapeutic strategies, focusing on the purinergic system.

The review is interesting and topical, as SARS-Cov-2 infection has triggered the progression of several chronic diseases, including neurodegenerative diseases.

There are several points that are important to address in this review to do it as comprehensive as possible.

1.         It is important that a table be made of the studies that have been conducted between ALS and COVID-19.

2. To point out the studies in humans on the expression of P2X1 to P2X14 receptors as well as P2Y with ALS.

3. To point out the drugs used for modulation of purinergic pathways.

4.         To discuss in depth the interaction of SARS-CoV-2 on purinergic receptor expression, point out studies that have been performed in primary source articles (not reviews).

Reviewer 3 Report

Comments and Suggestions for Authors

The authors presented a nice and well-written narrative review, highlighting the current evidences related to SARS-CoV-2 infection, COVID-19 and ALS. Despite the present literature being limited in its descriptions of several cases and, thus, providing few evidences the authors have properly discussed all the important aspects associated with neuroinfectious and neuroinflammatory mechanisms and the development of neurodegenerative disorders, such as ALS. Several previous infectious diseases have been previously associated as potential triggers for Motor Neuron Disease/ALS, such as HIV, enterovirus and EBV. One important aspect is that few of these patients have been properly investigated regarding the existence of a possible monogenic and oligogenic basis both in sporadic and familial contexts. The graphical abstract has been nicely represented by the authors. It perfectly summarizes the main ideas and concepts discussed in the text in an easy way for the general physician, neurologist and basic area researcher. Certainly, the follow-up of patients during the next years after COVID-19 pandemics end will enable a correct interpretation of the real meaning of COVID-19 infection as potential trigger of ALS. The authors emphasized the importance of purinergic pathway changes and P2X7, P2X4, and A2AR receptors in the pathogenesis of ALS and possible correlation with dysfunctions related to the viral effect. I only ask the authors if it would be possible to mention if any of the currently approved drugs for ALS treatment have any direct effect in purinergic receptors or pathways, including Riluzole, Edaravone, and Tauroursodiol with Sodium phenylbutyrate. In the case of any effect due to mechanism of action, it would be interesting to add this aspect in the Discussion. 

Round 2

Reviewer 1 Report

Comments and Suggestions for Authors

The Authors have improved their manuscript as requested.

Comments on the Quality of English Language

Minor proofreading 

Reviewer 2 Report

Comments and Suggestions for Authors

Thanks to the authors for their replies.

The authors did important changes, so the article improved considerably.